# EEG-EyeTrack: A Benchmark for Time Series and Functional Data Analysis with Open Challenges and Baselines

## Abstract

A new benchmark dataset for functional data analysis (FDA) is presented, focusing on the reconstruction of eye movements from EEG data. The contribution is twofold: first, open challenges and evaluation metrics tailored to FDA applications are proposed. Second, functional neural networks are used to establish baseline results for the primary regression task of reconstructing eye movements from EEG signals. Baseline results are reported for the new dataset, based on consumer-grade hardware, and the EEGEyeNet dataset, based on research-grade hardware.

## 1 Introduction

In the future, brain-computer interfaces (BCIs) might offer the possibility of restoring or augmenting sensory perception, such as allowing a blind person to see (Muqit et al., 2024), or even converting thoughts into text (Willett et al., 2023). Currently, electroencephalography (EEG), the technology behind modern BCIs, is used primarily to assist in the diagnosis of neurological diseases (Behzad & Behzad, 2021; Britton et al., 2016). Automated analysis of EEG data is an active area of research. However, there is still a long way to go before brain activity can be reliably analyzed, and used for the development of robust BCIs. A more feasible application of BCIs is EEG-based eye tracking Kastrati et al. (2021); Fuhl et al. (2023); Dietrich et al. (2017); Sun et al. (2023). Typically, electric activity related to eye movements is ignored or filtered out (Croft & Barry, 2000), but it might be used to reconstruct the gaze direction. A reliable and accurate tracking of eye movement opens up new possibilities for BCIs. EEG-based eye tracking has the advantage of requiring no additional hardware when brain activity is already being monitored with an EEG headset. Additionally, it remains effective even when the eyes are closed, such as during sleep. However, current EEG-based eye trackers are mainly based on EEG data recorded with expensive hardware in a laboratory setting. Open datasets recorded with consumer-level hardware outside a lab environment are scarce, yet crucial for the development of EEG-based eye trackers that work reliably in precisely this context.

Two promising fields of statistics for the interpretation of EEG data are time series and functional data analysis (FDA). FDA provides statistical tools for data that may be modelled as smooth functions (of time, space, frequency or any other reasonable argument), see Ramsay & Silverman (2005) and Kokoszka & Reimherr (2017) for foundational overviews. Recent extensions into deep learning give rise to functional neural networks (FNNs), which embed such smoothness directly into network layers (see Rossi et al., 2002; Rao & Reimherr, 2023b; Heinrichs et al., 2023, among others). It can be assumed that developments in these areas will enable robust and accurate eye tracking. However, baseline datasets for functional data are often either too simple or nearly impossible to analyze with modern methods. The aim of this work is two-fold. On the one hand, the Consumer-Grade EEG-Based Eye Tracking Dataset (Afonso & Heinrichs, 2025) is introduced as a new, challenging benchmark for methods in the areas of time series analysis and FDA. For this purpose, open problems and evaluation metrics are established. On the other hand, the use of methods from FDA is investigated for the central problem of eye movement reconstruction from EEG data, which can be considered a scalar-on-function or function-on-function regression problem, and first baseline results are given.

To obtain the baseline results, an existing model for eye movement reconstruction, as proposed by Fuhl et al. (2023), was used as a reference. Recently, neural networks with functional layers were specifically designed for the analysis of EEG data. These functional neural networks (FNNs) are expected to better model the smoothness of the data, and therefore require fewer parameters and be more robust to noise compared to other models. As FNNs have not been used for EEG-based eye tracking, the models are also compared based on the already existing EEGEyeNet dataset.

The majority of FDA methods requires previous alignment of curves, which is referred to as *function registration*. However, this is only possible if meaningful features can be identified in the data or if events of interest are externally triggered. In the case of EEG data, where the event of interest is often a self-paced action and the signal is noisy and complex, curve registration is infeasible. Therefore, EEG-based eye tracking data may be of interest for developing and testing new methods that work with unregistered data. However, due to the experimental design, the true eye movement data is known and can serve as a basis for curve alignment. This allows methods that require function registration to be applied to the dataset as well.

To summarize, our contribution is as follows:

- **Standardized FDA benchmark:** We formalize scalar-on-function regression on the public "Consumer-Grade EEG-Based Eye Tracking Dataset" with standardized evaluation metrics for reproducible FDA benchmarking.

- **Baseline performance:** We provide initial benchmarks by evaluating different functional and conventional models, on both the consumer-grade dataset and the research-grade EEGEyeNet, establishing reference numbers.

## 2 Related Work

Typically, when recording EEG data, one is interested in brain activity rather than artifacts from muscle or eye movements. The latter are either filtered out, with methods such as independent component analysis (ICA) or canonical correlation analysis (CCA), or completely ignored, especially in deep learning-based pipelines (Urigüen & Garcia-Zapirain, 2015). Recently, however, the reconstruction of eye movements from EEG recordings has been studied in the literature as an independent task. Dietrich et al. (2017) recorded short segments of EEG data, containing extreme eye movements (left, right, up, down), with an EEG headset with 14 wet electrodes. They classified the recordings with a variant of $k$-nearest neighbors, reaching an accuracy of 96%. Subsequently, Sun et al. (2023) recorded eye movements in a laboratory setting with 64 wet electrodes. The proposed algorithm, EEG-VET, was able to reconstruct *saccadic* (rapid) and smooth eye movements. Recently, the EEGEyeNet dataset was introduced as a benchmark dataset for the problem of reconstructing eye movement from EEG recordings (Kastrati et al., 2021). The dataset contains recordings from 356 subjects, comprising 38 hours of saccadic left-right eye movements, 7 hours and 52 minutes of saccadic movements on a grid, and 1 hour and 29 minutes of free eye movements. The EEG data was recorded in a laboratory setting with an EEG headset containing 128 electrodes. Based on the EEGEyeNet, Fuhl et al. (2023) proposed a neural network architecture, where the first layer should act as a (learnable) spatial filter. This model, referred to as *SpatialFilterCNN* in the remainder of this work, led to new state-of-the-art results for the EEGEyeNet dataset and is used as benchmark for subsequent experiments. More recently, Afonso & Heinrichs (2025) introduced a new dataset, similar to EEGEyeNet, yet recorded with consumer-grade hardware outside a controlled laboratory setting, that shall allow the development of EEG-based eye trackers in real-world applications. The used EEG-headset had only four dry electrodes, and therefore a substantially lower signal-to-noise ratio. We will refer to the dataset as "Consumer-Grade EEG-based Eye Tracking" dataset.

Functional data analysis (FDA), a field of statistics, deals with smooth processes. If eye movements are recorded with a sufficiently high sampling frequency, the resulting data can be regarded as a smooth function of time. A variety of open datasets for different tasks are commonly used as benchmarks. These datasets include the "Berkeley Growth Study" (Tuddenham, 1954), daily temperatures from Canadian weather stations and the "Handwriting" dataset (Ramsay & Silverman, 2005). For classification of functional data, the

"Phoneme" and "Tecator" datasets are frequently used (Hastie et al., 1995; Thodberg, 2015). Especially for the latter two, very high accuracies ($> 91\%$ and $100\%$, respectively) have been achieved in the literature (Heinrichs et al., 2023). For future developments in FDA, new and challenging datasets are required. Based on the "Consumer-Grade EEG-based Eye Tracking" dataset, we formulate multiple open challenges in FDA in Section 4.1.

Most FDA methods require data to be *registered*, which means aligning the (functional) data to a common time axis, where each time point contains essentially the same information across all functions (Kneip & Gasser, 1992; Gasser & Kneip, 1995; Ramsay & Li, 1998). This does not only include classic methods, such as functional PCA (FPCA) and functional linear models (Shang, 2014; Cardot et al., 1999; Cuevas et al., 2002), but also modern methods for functional time series, such as tests on stationarity or white noise (Bücher et al., 2020; 2023).

Although curve registration is a common preprocessing step, it is not feasible in many applications, especially when sliding windows are considered. Therefore, different shift-invariant methods have been proposed recently, such as transform-invariant FPCA (Heinrichs, 2024). For classification and regression tasks, functional neural networks have been proposed and extensively studied (Rossi et al., 2002; Rossi & Conan-Guez, 2005; Rossi et al., 2005). Early functional neural networks essentially project the (infinite-dimensional) functions onto multivariate vectors in the first layer and use "standard" layers throughout the remainder. More recently, fully functional neural networks have been proposed for scalar-on-function and function-on-function regression (Rao & Reimherr, 2023a;b). Additionally, convolutional layers have been extended to functional data (Heinrichs et al., 2023). In the remainder, we will use the definitions of functional neurons from the latter reference, as it has been applied to EEG data in the past, and refer to it for explanations of the functional layers and their hyperparameters.

## 3    Functional Data Analysis

Functional data analysis (FDA) offers a framework for modeling signals, such as EEG recordings, as smooth functions $x(t)$ rather than high-dimensional vectors of discrete time points. With this approach, dependencies between neighboring points are explicitly taking into account. A first step in FDA, usually consists of smoothing the discrete observations to obtain smooth data. Afterwards, the functional data is often projected from an infinite-dimensional function space to a lower dimensional space by dimension reduction techniques, such as functional principal component analysis (FPCA), so that multivariate methods can be used for a subsequent analysis.

Through dimension reduction, important information might get lost, and it can be advantageous to work directly with functional data. Depending on the analysis' goal, it is often assumed that the observations are elements of $C([0, 1])$, the Banach space of continuous functions on $[0, 1]$, or elements of $L^2([0, 1])$, the Hilbert space of square-integrable functions on $[0, 1]$.

Let $H$ denote the function space of the data. A central problem of FDA, is the task of regression, that links a functional input to a scalar or functional response. In *scalar-on-function* regression, a scalar outcome $y$ is modeled as $y = F(x) + \epsilon$, where $F : H \to \mathbb{R}$ denotes a (not necessarily linear) functional from the function space $H$ to $\mathbb{R}$. When $F$ is assumed to be linear, we obtain the linear model

$$y = \alpha + \int \beta(t)x(t)\mathrm{d}t + \epsilon,$$

where $\beta(t)$ is a coefficient function. *Function-on-function* regression generalizes this to predict an entire output curve $y(t)$, e.g., a gaze trajectory. In this case, $y = F(x) + \epsilon$, where $F : H \to H$ denotes an operator. Again, if $F$ is linear, we obtain the linear model

$$y(t) = \alpha(t) + \int \beta(t, s)x(s)\mathrm{d}s + \epsilon(t).$$

Functional neural networks (FNNs) allow modelling of non-linear relations between input and output. In these networks, discrete sums and convolutions are essentially replaced by their continuous counterparts.

For input functions $x_1, \ldots, x_p$, defined on $[0, 1]$, a functional neuron might be defined by

$$y(t) = \sigma\left(w_0(t) + \sum_{i=1}^{p} w_i(t)x_i(t)\right),$$

for functional weights $w_0, \ldots, w_p$. Further, we may extend $x_1, \ldots, x_p$ to $[-b, 1+b]$, for some bandwidth $b > 0$, by defining the functions as zero in $[-b, 0) \cup (1, 1+b]$. Then, we can define a functional convolutional layer by

$$y(t) = \sigma\left(w_0(t) + \sum_{i=1}^{p} \int_{-b}^{b} w_i(s)x_i(t-s)\mathrm{d}s\right),$$

for functional weights $w_0 : [0, 1] \to \mathbb{R}$ and $w_1, \ldots, w_p : [-b, b] \to \mathbb{R}$, see Heinrichs et al. (2023). In the following, we consider smooth (continuous or differentiable) weights $w$. Due to the integration, the output becomes increasingly smooth. Generally, for differentiable weight functions $w_1, \ldots, w_p \in C^1([0, 1])$ and $k$-times differentiable inputs $x_1, \ldots, x_p \in C^k([0, 1])$, $y$ will be $k+1$ times differentiable. Moreover, for bounded functions $x_1, \ldots, x_p$, $y$ is continuous and for continuous functions $x_1, \ldots, x_p$, $y$ is differentiable. This observation is crucial for the use of functional neural networks, and explains how deep FNNs filter out local noise.

Beyond regression, FDA encompasses tasks such as classification, clustering curves by shape, and detecting change points in functional time series. In this paper, we focus empirically on scalar-on-function regression, while defining open challenges related to other FDA tasks.

## 4 Dataset and Open Challenges

We considered the "Consumer-Grade EEG-based Eye Tracking" dataset, which is publicly available on Zenodo[1], originally published by Afonso & Heinrichs (2025). The dataset contains recordings from 116 sessions of 113 participants. Each session lasted approximately 6 minutes, yielding a total of 11 hours and 45 minutes.

The experiments consisted of a target moving on screen that participants were asked to follow closely. Data from three different modalities was recorded. First, EEG-data was measured at positions TP9, TP10, AF7, AF8 according to the international 10-10 system, where the electrode Fpz was used as reference. Second, the current gaze position, as estimated from a webcam-based eye tracker, was recorded. And finally, the target's position on screen was tracked.

Each session consisted of four stages. In the first two stages ("level-1" recordings), the target moved only horizontally and vertically on screen, while in the latter two stages ("level-2" recordings), the target moved in more directions, yielding more degrees of freedom. Each level had one stage, where the target changed its position abruptly, and another stage, where the target moved continuously across the screen. The former recordings are referred to as "saccades" while the latter as "smooth". Especially, recordings with smooth eye movements are of interest as benchmark for methods in functional data analysis.

Missing values in the data streams, that occurred due to hardware issues, were imputed through a suitable SARIMA model. For an explanation of this process, as well as details regarding the experiments and data acquisition, we refer to the original Data Descriptor (Afonso & Heinrichs, 2025). For our experiments, we used the preprocessing as described therein. More specifically, notch filters at $50\,\mathrm{Hz}$ and $60\,\mathrm{Hz}$ were used, and subsequently a bandpass filter between $0.5\,\mathrm{Hz}$ and $40\,\mathrm{Hz}$. Note that the transition from the stopband to the passband of the bandpass filter allows some high-frequency noise to persist. This is mitigated by applying the notch filters beforehand. Additionally, we excluded recordings with known quality issues (recordings from participants 2, 4, 16-20, 62-67, 79 and the "level-1-saccades" recording from participant 50).

Across all 4 EEG channels and 407 recordings, missing samples due to hardware dropouts ranged from $0.0\,\%$ to $24.1\,\%$ per channel (median $6.3\,\%$), with 27 recordings exceeding 10% of missing data (see Table 1). To preserve both non-stationary trends and the strong $50\,\mathrm{Hz}$ background noise, Afonso & Heinrichs

---

[1] https://zenodo.org/records/14860668

(2025) employed a SARIMA model to fill in missing values. For compatibility with other works based on the original data, we chose to use work with the preprocessed data, where missing values have been imputed with a SARIMA model with seasonal lag 5 and the remaining parameters tuned based on the Akaike Information Criterion.

| Statistic | Value (%) |
|---|---|
| Minimum missing per channel | 0.0 |
| Median missing per channel | 6.3 |
| Maximum missing per channel | 24.1 |
| Recordings with >10% missing | 27 |

Table 1: Summary of missing-data rates across EEG channels and recordings.

For details regarding participant demographics, recording environment, ethics/consent procedures, licensing conditions, and additional information, we refer to the original Data Descriptor (Afonso & Heinrichs, 2025). While this study does not involve new data collection, the dataset used contains EEG signals that could potentially be misused for applications such as covert attention tracking or surveillance. Although the dataset itself does not include any personal identifiers, we caution against such misuse. We encourage readers to refer to the original data descriptor for the ethical considerations.

### 4.1 Open Challenges

Due to the experimental design, the dataset is well suited to serve as a benchmark for FDA methods. The main challenge of this dataset is the prediction of the target's position from the EEG signal. This can be formulated as a **function-on-function** or as a **scalar-on-function regression** problem, where in the first case the position over time and in the latter case the final position of the target should be predicted from the EEG data. Note that instead of the target's position, the gaze position, as estimated from a webcam-based eye tracker, might be used as a target variable as well.

The dataset can be used as a benchmark for FDA methods that require function registration, as well as for methods that do not require it. In level-1 experiments, the target starts in the center of the screen and moves up, down, left or right. The start of each movement can be used as a marker to divide an entire recording into multiple trials. These trials can be assumed to be aligned curves (or further registered). For methods that do not require registration, sliding windows can be generated from the entire recording.

For benchmark experiments, the originally proposed train-test split (90% training, 10% test) should be kept, and test data from other tasks/levels should not be used for training. For regression, both coordinates of the target, as defined in the columns `Stimulus_x` and `Stimulus_y` should be predicted. As a metric to measure the quality of predictions, we propose the Mean Euclidean Distance (MED) between the predicted and the ground truth stimulus position. The MED is defined as

$$\text{MED}_k = \frac{1}{N_k} \sum_{i=1}^{N_k} \sqrt{(x_i - \hat{x}_i)^2 + (y_i - \hat{y}_i)^2},$$

where $N_k$ denotes the number of samples in the $k$-th recording of the test data, $x_i$ and $y_i$ are the true, and $\hat{x}_i$ and $\hat{y}_i$ are the predicted target positions at time step $i$. The final score of a model is the MED over all 12 recordings in the test data, which is defined as

$$\text{MED} = \frac{1}{\sum_{k=1}^{12} N_k} \sum_{k=1}^{12} N_k \text{MED}_k.$$

The MED is a standard metric in eye tracking, because of its easy and clear interpretation (Raghunath et al., 2012; Papoutsaki et al., 2018; Dalrymple et al., 2018). It measures the average distance between the ground truth stimulus position and its corresponding prediction. Other possible metrics would have been the MSE,

RMSE, or MAE, but compared to the MED they do not result in values that are easily interpretable, as they correspond to the mean of their respective one-dimensional metrics for the $x$- and $y$-axis.

When reporting the performance of a model, the MED for each task should be reported. This allows for a more nuanced comparison of the performance of different methods, as models could overfit to one task and perform poorly on others, making them unsuitable for general purpose eye tracking.

A schematic overview of the benchmarking pipeline is given in Figure 1 (a).

Besides the main challenge of (scalar-on-function) regression, we identified several additional challenges:

1. **Classification of Movements:** Classify EEG data based on eye movements, e. g., "horizontal" vs. "vertical", "saccades" vs. "smooth", or "up"/"down"/"left"/"right".

2. **Classification of Participants:** Classify participants into different groups with class labels generated from eye movements, e. g., as good or poor trackers, based on the difference between target and gaze position; or as fast or slow trackers, based on the lag between target and gaze position.

3. **Clustering:** Identify brain activity patterns from EEG signals.

4. **Dimension Reduction:** Reduce the dimension of the data while minimizing the reconstruction loss.

5. **Outlier detection:** Identify segments, or time points, with unusual data, e. g. due to missing values that have been replaced by zeros or other erroneous measurements.

6. **Change point detection:** Detect moments where gaze tracking or EEG activity shifts significantly, e. g., due to a loss of attention or transitions between movements and pauses.

## 5 Model Architectures

Because FNNs are rather new and effective design practices are not yet established, we relied on established CNN architectures as a reference point to guide the development of our FNN architectures. These architectures typically consist of three main components: the *stem*, *body*, and *head*.

**Stem:** The stem is the initial part of the network responsible for converting the input signal into a form that can be processed by subsequent layers. In a traditional CNN architecture, the stem often includes a convolutional layer with a large kernel size, followed by a pooling layer to reduce the spatial dimensions of the input. For our FNN, we replaced the standard convolutional layer with a functional convolutional layer, using a large resolution to mimic the large kernel size of conventional CNN. In addition, we placed a spatial filtering layer in front of this layer, inspired by early experiments with the SpatialFilterCNN, where it was found to be beneficial for enhancing the model's performance. This design formed the foundation of the stem used in all the architectures we explored. The general structure of the stem is shown in Table 6 of Appendix A.

**Body:** The body of the network is where the bulk of the computation takes place. It is composed of multiple stages, each consisting of several blocks. For two-dimensional signals (like images) a typical block is structured as a sandwich of three convolutional layers, where the outer two use $1 \times 1$ kernels, and the middle layer uses a $3 \times 3$ kernel. In our architectures, we decided to omit the first $1 \times 1$ convolution following insights from the SpatialFilterCNN. Further, we set the kernel length of each filter to 9. Each convolutional layer was followed by batch normalization before applying the activation function, which is a common practice. Additionally, we incorporated residual connections within each block, allowing the input to be added to the output of the last convolutional layer. The resulting block structure is shown in Table 7 of Appendix A.

As in CNNs, each stage of our FNN body concludes with a pooling layer, that reduces the length of the input signal by a factor of two, effectively narrowing the signal as it progresses deeper into

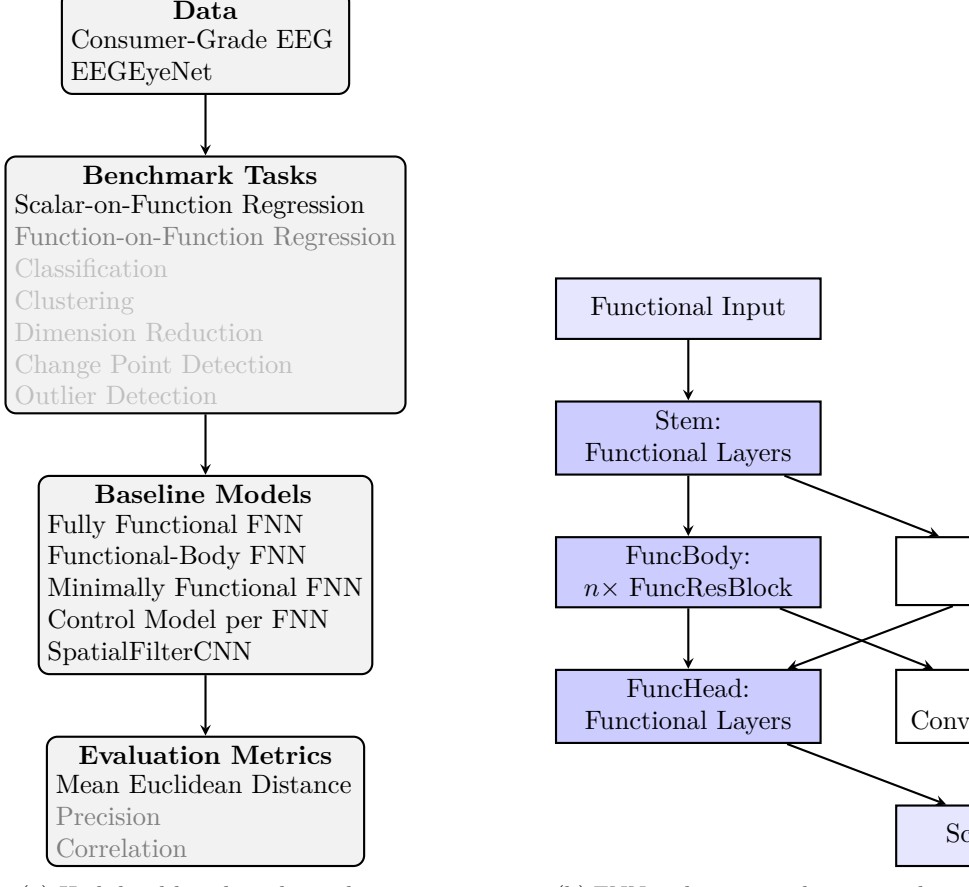

(a) High-level benchmark pipeline.

(b) FNN architectures decomposed into Stem, Body, and Head. Functional layers shaded blue; input/output layers shaded in light blue; conventional layers left white.

Figure 1: (a) Benchmark pipeline from datasets through tasks to evaluation. (b) Functional neural network variants: every FNN has a stem (functional), a body (functional or conventional), and a head (functional or conventional).

the network. Concurrently, the number of filters in the convolutional layers was increased at each stage, resulting in a deeper network.

**Head:** In a typical CNN, the head consists of one or more dense layers that transform the output of the body into the final prediction. To accommodate this, the output from the body must first be flattened or aggregated (*e.g.* through global average pooling).

Based on this reference architecture, we designed three FNNs and evaluated their performance. We did this by replacing different parts of the architecture with functional layers. An overview of the model architectures is provided in Figure 1 (b), while additional information are provided in the appendix. An implementation of the models under MIT license is available online: link to non-anonymous repository

All architectures are sized to have a similar number of parameters of around $1.2 \cdot 10^6$, resulting in models of approximately 4 MB in size. This way, the models are comparable in terms of complexity.

## 5.1 Fully Functional Architecture

The first FNN follows a "fully functional" design, meaning that every component in the network is functional. For this, the residual block from the reference architecture is replaced by a "functional residual block", where

the first convolution is changed to a functional convolutional layer. The second convolution, which uses a kernel size of 1, remains unchanged, as a functional convolutional layer with resolution 1 coincides with a standard convolutional layer. The functional residual block is displayed together with the standard residual block in Table 7 of Appendix A. Parts that differ between the two blocks are marked accordingly.

Multiple of such functional residual blocks are then chained together, making up one big stage, with the number of filters increasing progressively with the depth of the network. No pooling layers are used in or after the stage. There are two reasons for this: First, pooling operations with a stride break the smooth structure of the signals passing through the network. Second, using only functional layers at the head avoids the "parameter explosion" that would occur after the flattening operation.

In the head of the network, solely functional dense layers are employed. Because these layers expect functional inputs, there is no need to flatten the output of the body. Instead, the body's output is fed directly to the head. This leads to a significant reduction in the number of parameters. Without the need to flatten the body's output, the first dense layer in the head requires only `last_channels_out` $\times$ `neurons` weights. In contrast, if the output had been flattened, the number of weights would be `last_channels_out` $\times$ `last_steps` $\times$ `neurons`, where `last_steps` refers to the number of time steps remaining after the body.

For instance, with a window size of 512 and no pooling, the `last_steps` would be 512. Assuming `last_channels_out` is 256 and there are 256 neurons in the first dense layer, this requires only $256 \times 256 = 65,536$ weights, which equates to approximately $257\,\text{KB}$, assuming 4-byte floats. In contrast, if the output had been flattened, the network would require $512 \times 256 \times 256 = 33,554,432$ weights, or approximately $128\,\text{MB}$ with 4-byte floats.

The complete architecture of the first "fully functional" FNN is displayed in Table 8 of Appendix A.

## 5.2 Functional Body Architecture

The second FNN architecture takes a hybrid approach, using functional layers only in the body while concluding with standard dense layers in the head. Unlike the fully functional architecture, using standard dense layers in the head necessitates pooling layers at the end of each stage, to reduce the number of steps flowing into the head.

The body of this architecture is structured into two stages, each composed of two functional residual blocks. In the first stage, each block contains 64 filters, while in the second stage, the number of filters is increased to 112. Pooling operations at the end of each stage reduce the input size by half, helping to control the dimensionality of the data as it passes through the network.

The output of the body is then flattened and passed through the head, which consisted of two standard dense layers. The first dense layer contains 64 neurons with an ELU activation function, while the second layer has two neurons with a linear activation function, producing the final output. Table 9 of Appendix A provides a detailed breakdown of the second FNN architecture.

## 5.3 Minimally Functional Architecture

The third FNN architecture uses functional layers sparingly, with only a single functional dense layer in the head and the functional convolutional layer in the stem, that is part of all three architectures. The former functional layer helps avoid the need to flatten the body's output, which similar to the first architecture, reduces the number of required parameters. The "saved" parameters are reallocated to an additional larger dense layer in the head, rather than adding more blocks to the body. This approach introduces some architectural variety. The body of this architecture is similar to that of the second FNN, but it employs standard residual blocks rather than functional ones. At the head, the output from the body is first aggregated by a functional dense layer with 512 neurons. This layer includes pooling and uses the ELU activation function. It acts as a more flexible global average pooling layer, providing the flexibility to weigh different parts of the signal differently, as opposed to standard global average pooling, which averages all inputs equally. After the functional layer, the data is processed by a standard dense layer with 512 neurons, also using the ELU activation function. The architecture concludes with a final standard dense layer with two neurons and a

linear activation function for the output. The details of the third FNN architecture are shown in Table 10 of Appendix A.

## 6 Experiments

### 6.1 Metrics

Three metrics were used to evaluate model performance in the experiments: the Mean Euclidean Distance (MED), as described in Section 4.1, the Pearson correlation coefficient between true and predicted trajectories, and the precision. These metrics provide complementary insights into different aspects of model accuracy, helping us assess the quality of predictions across both smooth pursuit and saccadic eye movements.

Note that the term "precision", contrary to its use in classification, refers to a measure of variation of the predicted gaze position. More precisely, we define the precision as the MED between true and predicted direction of eye movement, i. e.,

$$\text{Precision} := \frac{1}{N} \sum_{i=1}^{N} \|\hat{y}_{i+1} - \hat{y}_i - (y_{i+1} - y_i)\|_2 \,, \tag{1}$$

where $y_i$ and $\hat{y}_i$ denote the true and predicted gaze position at time step $i$.

Since the Pearson correlation can only be computed for one-dimensional variables, we calculate the correlation separately for the $x$ and $y$ components of the predicted gaze positions. These separate metrics are referred to as $\text{corr}_x$ and $\text{corr}_y$, respectively. To obtain a single combined correlation metric, which can be used for tasks such as hyperparameter tuning, we compute the mean of these metrics.

While the MED serves as a primary measure of model performance, we introduced the Pearson correlation coefficient to complement the MED. It can, for example, detect models that predict constant values, such as the mean, as it will be (approximately) zero in this case.

Further, the Pearson correlation is translation and scale invariant. Models that predict generally the right direction, but not the correct position or scale, would be penalized by the MED, but will have a high correlation. This property makes the correlation especially valuable in early stages of model development, where capturing the direction is a positive sign of learning.

It is important to note, however, that while the correlation can provide valuable insights during model training, it is not as useful for benchmarking more advanced models. As models become more capable, they should not only capture the form but also accurately predict the scale and magnitude of the values. In these cases, MED becomes more relevant as a final evaluation metric, since it directly measures how close the predictions are to the true values.

### 6.2 Experimental Setup

The experiments were conducted on an NVIDIA DGX Workstation, the hardware specifications of which are outlined in Table 2. The system features four NVIDIA Tesla® V100-DGXS GPUs, each with 32 GB of memory, supported by an Intel Xeon E5-2698 v4 CPU running at 2.2 GHz with 20 physical cores, and 40 virtual cores. Additionally, the workstation is equipped with 256 GB of RAM.

While the workstation has the capacity to run multiple GPUs simultaneously, initial tests revealed that, using multiple GPUs did not result in a significant reduction in training time. This is likely due to the bottleneck in data loading. As a result, each experiment was conducted on a single GPU. However, multiple GPUs were used to run several experiments concurrently.

All experiments were conducted within a Docker container environment. Specifically, we utilized the NVIDIA TensorFlow container image `nvcr.io/nvidia/tensorflow:24.06-tf2-py3`, which provided an optimized runtime for TensorFlow with CUDA 11.4 support.

Hyperparameter tuning was handled using *Optuna*, with the TPE sampler configured for efficient search through the hyperparameter space.

| Component | Specification |
|-----------|---------------|
| GPUs | 4 × NVIDIA Tesla® V100-DGXS-32GB |
| CPU | 1 × Intel Xeon E5-2698 v4 2.2 GHz (20-Core/40 vCores) |
| RAM | 256 GB ECC Registered-DIMM DDR4 SDRAM |
| OS | DGX OS 5.4.2 (Ubuntu 20.04) |
| CUDA | 11.4 |

Table 2: Hardware specifications of the NVIDIA DGX Workstation used for the experiments.

# 7 Results and Discussion

## 7.1 Consumer-Grade EEG Eye Tracking

In the following, we compare the EEG-based reconstructions of eye movements with different baselines. The first baseline predicts the target's position randomly, and is used to verify if the model learns anything at all. As a stronger baseline, we use the mean position over the entire training set, which corresponds to the center of the screen. While this baseline is still trivial, it helps to check if the model learned something non-trivial. Finally, we use the webcam-based predictions as a strong baseline. This baseline is generally expected to perform well, up to some delay between movement and predictions, and is used for a direct comparison between EEG- and webcam-based eye tracking.

We initially used the SpatialFilterCNN with its original hyperparameters, i.e., $N_S = 16$, $N_1 = 32$, $N_2 = 64$, `spatial_filtering` enabled, and `equally_sized` convolutional layers. While the original paper used a window size of 500, we opted for 512 samples to align with our sampling rate of 256 Hz, resulting in 2-second windows. Further, we used both filtered and unfiltered data. Details on the architecture of the SpatialFilterCNN are provided in Appendix B.

The results suggested that filtered data improves predictions for the "saccades" paradigm, while unfiltered data is better suited for smooth eye movements. Additionally, we conducted hyperparameter tuning experiments, applying the previously identified optimal filtering — filtered data for "saccades" and unfiltered data for "smooth" tasks. Using Optuna, the hyperparameters were tuned by training 20 models for 30 epochs with different hyperparameter configurations, for each of the four tasks. The ranges of the tuned hyperparameters are provided in Table 11 in the Appendix.

We used recordings of participants 1, 20, 28, 42, 52, and 69 as validation data. These recordings were selected as a representative subset from the training data. After hyperparameter tuning, the final model was trained on the entire training dataset.

The functional neural networks were trained exactly as the SpatialFilterCNN, i.e., by using the Adam optimizer with a learning rate of 0.0008 and a batch size of 384 for 30 epochs, minimizing the mean-squared error. We did not adjust the hyperparameters of the FNNs to see how well they work out of the box for the given scalar-on-function problem. To evaluate the use of functional layers, we performed an ablation study. For this, we used control models for each architecture, where the functional layers were replaced by conventional layers. More specifically, for the fully functional architecture, the functional residual blocks were replaced by standard residual blocks and the two final functional dense layers were substituted by two convolutional layers followed by a global average pooling layer. The first convolutional layer contained 256 filters with ELU activation, and the second contained 2 filters with a linear (no) activation. Both layers had a kernel size of 12. For the functional body, the functional residual blocks were replaced by standard residual blocks. Finally, for the minimally functional neural network, we replaced the single functional dense pooling layer with a standard convolutional layer, a global average pooling layer, and an ELU activation. The convolutional layer had 512 filters, to match the 512 neurons of the functional dense layer, and a kernel size of 12. Global average pooling was used to obtain a scalar output, corresponding to the output of the functional dense pooling layer.

The results of the level-1 and level-2 experiments are given in Tables 3 and 4, respectively. Additionally, exemplary predictions of different neural networks are displayed in Figure 2.

Overall, all models made better predictions than random guessing. For level-1 experiments, the MED of the mean baseline is comparable or even lower than that of the webcam-based predictions. This effect is mainly due to the fact that level-1 experiments only included horizontal and vertical movements and short pauses in the center of the screen between these movements. The "mean baseline", remained at the origin, and predicted (at least) one coordinate correctly as 0, while the webcam-based predictions vary across both axes. When taking the correlations $\text{corr}_x$ and $\text{corr}_y$ into account, we see that the webcam-based predictions contain useful information about the target's position, while the mean baseline does not. As expected, the model with tuned hyperparameters performed best among the SpatialFilterCNNs. The results of the FNNs must be assessed more carefully. For the minimally functional and the functional body neural networks, the functional layers each yield better results than their classic counterparts. For the fully functional neural network, the classic variant yields the overall best results. In general, the FNNs seem to have problems predicting the target's $y$-coordinate, but outperform the SpatialFilterCNN in terms of $\text{corr}_x$. Notably, when directly comparing the SpatialFilterCNNs with the FNNs, the precision of the latter is substantially smaller. This is due to the rather smooth trajectories that the FNNs predict, compared to the rather rough trajectories of the SpatialFilterCNNs. Note that small values of the precision, as defined in equation 1, are preferable. Thus, even without hyperparameter tuning, the FNNs perform equally well or even better than the SpatialFilterCNNs for smooth movements.

For level-2 experiments, which introduce more degrees of freedom, the results look different. First, the "mean baseline" has a substantially higher MED, which stems from the fact that now two non-zero coordinates must be predicted. Overall, the webcam-based predictions yield the lowest MED and the highest correlations $\text{corr}_x$ and $\text{corr}_y$. For the FNNs, the functional versions yield generally better predictions than their conventional counterparts. For smooth movements, the FNNs yield similar results as the tuned SpatialFilterCNN, yet with substantially lower values for the precision metric. For abrupt movements, the MEDs of nearly all FNNs are lower than those of the SpatialFilterCNNs with default hyperparameters.

Generally, the SpatialFilterCNN with tuned hyperparameters has the lowest prediction error in terms of the MED. For smooth movements, the FNNs yield similar (level-2) or higher (level-1) correlations compared to the SpatialFilterCNN. This suggests that the FNNs are better at learning movement directions, while the SpatialFilterCNN is better at learning the exact magnitude of movements. The substantially lower precision metric of the FNNs suggests that their predictions are smoother and more consistent compared to those of the SpatialFilterCNN. Overall, the neural networks with functional layers seem to outperform their conventional counterparts. However, it should be noted that the comparison is biased in favor of the SpatialFilterCNN, as the hyperparameters of the FNNs were not tuned.

## 7.2 EEGEyeNet

The previous comparison is based on data recorded by consumer-grade hardware. In addition, we compare FNNs with conventional CNNs based on the EEGEyeNet dataset, which was recorded under laboratory-controlled conditions and with research-grade EEG equipment with more electrodes and wet sensors. This allows an evaluation of FNNs under optimal conditions, compared to the real-world conditions from the previous section.

We used the same architectures and hyperparameters as before. Furthermore, we have added the EEGNet, a standard neural network for the analysis of EEG data, to the comparisons.

For training, the Adam optimizer with a learning rate of 0.0001 and a batch size of 64 was used. Further, we employed early stopping based on the validation loss, with patience of 20 epochs. All models were trained for a maximum of 50 epochs, with a window size of 500, as specified by Kastrati et al. (2021). In line with the literature, each model was trained 5 times. The results, including the mean and standard deviation of the MED and mean absolute error (MAE), are summarized in Table 5.

Except for the control model of the fully functional neural network, all functional architectures outperformed the SpatialFilterCNN in both the MED and MAE metrics, resulting in new state-of-the-art results on the

| level-1-saccades | | | level-1-smooth | | | |
|---|---|---|---|---|---|---|
| Model | MED | Precision | MED | Precision | $\text{corr}_x$ | $\text{corr}_y$ |
| Random | 165.9 | 177.1 | 143.6 | 177.4 | $-0.005$ | 0.002 |
| Mean | 83.58 | 0.650 | 51.98 | 0.364 | 0 | 0 |
| Webcam | 81.89 | 1.479 | 74.59 | 0.890 | 0.687 | 0.347 |
| SFCNN (unfiltered) | 76.60 | 14.07 | 62.44 | 12.72 | 0.226 | 0.038 |
| SFCNN (filtered) | 66.54 | 7.087 | 59.32 | 12.79 | 0.140 | $-0.021$ |
| SFCNN (tuned) | 54.62 | 4.648 | 57.92 | 6.093 | 0.199 | 0.192 |
| FullyFunc | **73.02** | 1.908 | 61.18 | 0.898 | 0.220 | **0.052** |
| FullyFunc (control) | 73.41 | **1.772** | **55.11** | **0.882** | **0.285** | 0.011 |
| FuncBody | **78.35** | **2.278** | **57.47** | **0.965** | **0.259** | 0.027 |
| FuncBody (control) | 78.85 | 2.366 | 64.69 | 1.100 | 0.220 | **0.046** |
| MinFunc | **64.72** | **1.865** | **56.18** | **0.901** | **0.168** | 0.070 |
| MinFunc (control) | 71.13 | 1.920 | 58.30 | 1.007 | 0.154 | **0.076** |

Table 3: Results of the baseline and reference models for level-1 experiments. The better result between functional and control model is highlighted in bold, and the best result for each metric in each category is underlined.

| level-2-saccades | | | level-2-smooth | | | |
|---|---|---|---|---|---|---|
| Model | MED | Precision | MED | Precision | $\text{corr}_x$ | $\text{corr}_y$ |
| Random | 199.8 | 177.3 | 164.5 | 177.5 | 0.002 | 0.002 |
| Mean | 162.3 | 0.591 | 104.5 | 0.420 | 0 | 0 |
| Webcam | 87.19 | 1.474 | 77.00 | 0.999 | 0.897 | 0.753 |
| SFCNN (unfiltered) | 146.7 | 53.69 | 119.3 | 48.92 | 0.322 | 0.069 |
| SFCNN (filtered) | 139.5 | 33.65 | 128.0 | 46.37 | 0.226 | 0.038 |
| SFCNN (tuned) | 109.0 | 6.454 | 99.38 | 20.84 | 0.434 | 0.157 |
| FullyFunc | **127.5** | **2.276** | **100.8** | **0.913** | **0.409** | **0.138** |
| FullyFunc (control) | 128.8 | 2.391 | 104.4 | 1.092 | 0.364 | 0.104 |
| FuncBody | **130.6** | 2.842 | **104.2** | **1.161** | **0.384** | 0.097 |
| FuncBody (control) | 135.8 | **2.747** | 105.2 | 1.176 | 0.378 | **0.146** |
| MinFunc | **129.7** | 2.602 | **101.5** | **1.181** | **0.411** | **0.159** |
| MinFunc (control) | 132.2 | **2.494** | 108.2 | 1.336 | 0.348 | 0.087 |

Table 4: Results of the baseline and reference models for level-2 experiments. The better result between functional and control model is highlighted in bold, and the best result for each metric in each category is underlined.

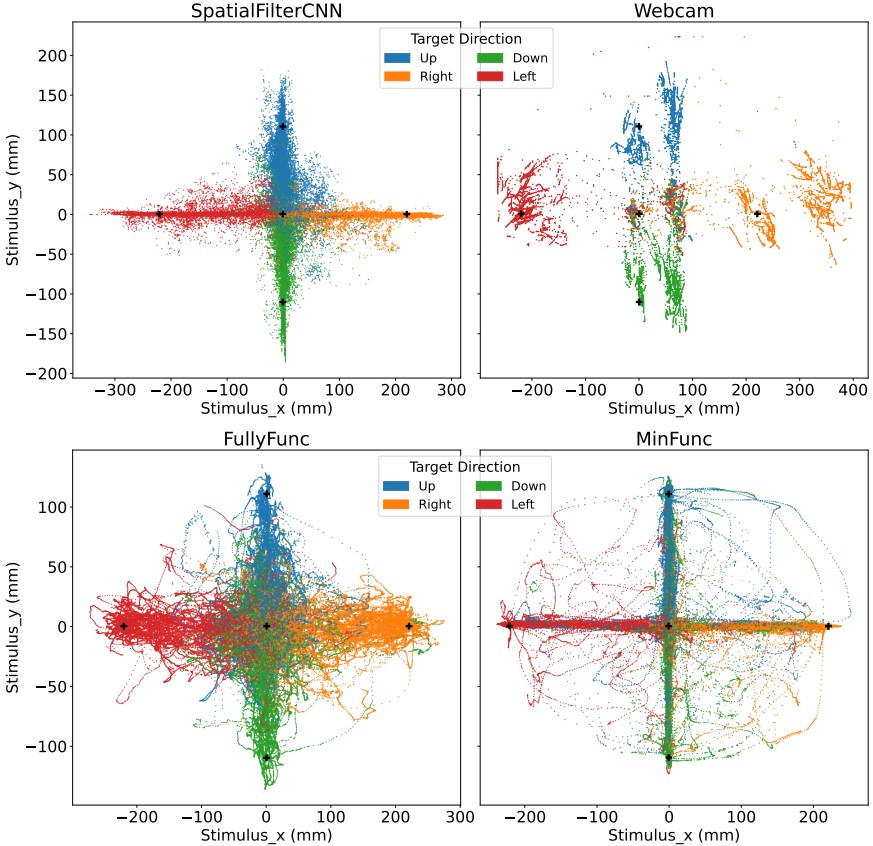

Figure 2: Comparison of various neural networks on the level-1 saccades task. Every prediction made on the test set is shown as a dot. The dots are colored based on the true target position. The four target positions are indicated by black crosses.

| Model | MED | MAE |
|---|---|---|
| FullyFunc | **68.5**($\pm$1.0) | **42.7**($\pm$0.6) |
| FullyFunc (control) | 69.9($\pm$1.3) | 43.6($\pm$0.8) |
| FuncBody | **68.0**($\pm$0.8) | 42.4($\pm$0.6) |
| FuncBody (control) | 68.1($\pm$0.5) | **42.3**($\pm$0.3) |
| MinFunc | 66.8($\pm$0.5) | 41.5($\pm$0.4) |
| MinFunc (control) | **66.2**($\pm$0.8) | **41.1**($\pm$0.5) |
| EEGNet | 77.3($\pm$0.3) | 48.7($\pm$0.2) |
| SpatialFilterCNN | 68.8($\pm$1.4) | 42.9($\pm$0.8) |

Table 5: Results of various models on the EEGEyeNet dataset. The mean (and standard deviation) of the MED and MAE are shown for each model. The better result between the functional and control model is highlighted in bold, and the best result for each metric is underlined. The MED and MAE are reported in millimeters using a conversion factor of 0.5.

EEGEyeNet benchmark. The best-performing model was the MinFunc control model, which surpassed the SpatialFilterCNN by 2.6 mm in the MED and 1.8 mm in the MAE.

It is important to note that the architectures used in these experiments were not tuned specifically for the EEGEyeNet dataset, and it is likely that further improvements could be achieved with hyperparameter tuning. Unlike the results from Section 7.1, where functional layers showed a clear benefit, the differences between the functional, and control models on the EEGEyeNet dataset were much smaller. In fact, the control models outperformed the functional models half of the time. Therefore, the advantage of functional layers in the constructed architectures is less clear when evaluated on the EEGEyeNet dataset.

Finally, we note that surpassing the current state of the art with a model architecture largely based on standard convolutional network architectures suggests that there is still significant room for improvement in the field of EEG-based eye tracking.

## 8 Conclusion

We introduced the "Consumer-Grade EEG-based Eye Tracking" dataset as a new benchmark for methods in the field of functional data analysis. More specifically, we stated a number of open challenges (classification, clustering, dimension reduction, outlier and change point detection) related to the dataset. We further studied the use of functional neural networks to solve the main challenge associated with the dataset, reconstructing a target's position on the screen from raw EEG data. This task can be formulated as function-on-function or scalar-on-function regression problem. We provided benchmark results for the latter. While our focus was on the main challenge, the remaining tasks outlined, such as classification and change point detection, offer valuable opportunities for further exploration. Baseline results for these tasks are yet to be established and provide a promising direction for future research.

While the results on the EEGEyeNet dataset were less conclusive, the results from Section 7.1 suggest that functional neural networks have beneficial properties for the prediction of the target's position. The ablation study carried out, in which functional and conventional neural networks were compared, generally showed better results for the respective functional version. For smooth movements, FNNs outperformed the non-tuned SpatialFilterCNNs, while achieving similar MED, $\text{corr}_x$ and $\text{corr}_y$ and substantially lower precision compared to the tuned SpatialFilterCNN. Note that with the definition of "precision" from equation 1, small values are better. The comparably small values for the precision are probably due to the fact that the predictions of FNNs tend to be rather smooth.

Overall, the results suggest that FNNs are a powerful tool for predicting smooth targets, which makes them particularly promising for analyzing functional data. Hyperparameter tuning was intentionally excluded to assess the capability of FNNs with naively chosen values. Future studies should focus on effective (and efficient) hyperparameter optimization for FNNs and best practices for default values.

In conclusion, the "Consumer-Grade EEG-based Eye Tracking" dataset allows a comparison of FDA methods across different open challenges. While FNNs have been found to be a helpful method for analyzing the dataset, it remains an open question how well other approaches work. It is expected that progress on this dataset will advance both EEG analysis and functional data analysis.

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

## A   Details on the Functional Neural Networks

| Layer | Parameters | Output Shape |
|-------|-----------|--------------|
| Input | — | $(\text{batch\_size}, \text{window\_size}, 4)$ |
| Conv1D | kernel_size: 1, filters: 16 | $(\text{batch\_size}, \text{window\_size}, 16)$ |
| BatchNorm | axis: -1 | $(\text{batch\_size}, \text{window\_size}, 16)$ |
| ReLU | — | $(\text{batch\_size}, \text{window\_size}, 16)$ |
| FuncConv1D | padding: same, resolution: 128, n_functions: 9, basis_type: Fourier | $(\text{batch\_size}, \text{window\_size}, 64)$ |
| AvgPool | pool_size: 2, strides: 2 | $(\text{batch\_size}, \text{window\_size}/2, 64)$ |

Table 6: The general structure of the stem, which is used by all three FNN architectures.

| Layer | Parameters | Output Shape |
|-------|-----------|--------------|
| Input | — | $(\text{batch\_size}, \text{steps}, \text{channels\_in})$ |
| (Func)Conv1D | padding: same, filters: channels_out (*standard only*) kernel_size: 9 (*functional only*) resolution: 24, n_functions: 6, basis_type: Legendre | $(\text{batch\_size}, \text{steps}, \text{channels\_out})$ |
| BatchNorm | axis: -1 | $(\text{batch\_size}, \text{steps}, \text{channels\_out})$ |
| elu | — | $(\text{batch\_size}, \text{steps}, \text{channels\_out})$ |
| Conv1D | padding: same, filters: channels_out, kernel_size: 1 | $(\text{batch\_size}, \text{steps}, \text{channels\_out})$ |
| BatchNorm | axis: -1 | $(\text{batch\_size}, \text{steps}, \text{channels\_out})$ |
| Add | — | $(\text{batch\_size}, \text{steps}, \text{channels\_out})$ |
| elu | — | $(\text{batch\_size}, \text{steps}, \text{channels\_out})$ |

Table 7: The structure of a standard ResBlock and a FuncResBlock.

**Architecture #1: Fully Functional**

| Layer | Parameters | Output Shape |
|-------|-----------|--------------|
| STEM | — | $(\text{batch\_size}, \text{window\_size}/2, 64)$ |
| FuncResBlock | filters: 64 | $(\text{batch\_size}, \text{window\_size}/2, 64)$ |
| FuncResBlock | filters: 96 | $(\text{batch\_size}, \text{window\_size}/2, 96)$ |
| FuncResBlock | filters: 144 | $(\text{batch\_size}, \text{window\_size}/2, 144)$ |
| FuncResBlock | filters: 216 | $(\text{batch\_size}, \text{window\_size}/2, 216)$ |
| FuncDense | neurons: 256, n_functions: 12, basis_type: Legendre, activation: elu | $(\text{batch\_size}, \text{window\_size}/2, 256)$ |
| FuncDense | neurons: 2, n_functions: 12, basis_type: Legendre, activation: linear, pooling: True | $(\text{batch\_size}, 2)$ |

Table 8: Architecture of the "fully functional" neural network. This model has $1,150,488$ trainable parameters, amounting to $4.39\,\text{MB}$.

**Architecture #2: Functional Body**

| Layer | Parameters | Output Shape |
|-------|-----------|--------------|
| STEM | — | $(\text{batch\_size}, \text{window\_size}/2, 64)$ |
| FuncResBlock | filters: 64 | $(\text{batch\_size}, \text{window\_size}/2, 64)$ |
| FuncResBlock | filters: 64 | $(\text{batch\_size}, \text{window\_size}/2, 64)$ |
| AvgPool | pool_size: 2, strides: 2 | $(\text{batch\_size}, \text{window\_size}/4, 64)$ |
| FuncResBlock | filters: 112 | $(\text{batch\_size}, \text{window\_size}/4, 112)$ |
| FuncResBlock | filters: 112 | $(\text{batch\_size}, \text{window\_size}/4, 112)$ |
| AvgPool | pool_size: 2, strides: 2 | $(\text{batch\_size}, \text{window\_size}/8, 112)$ |
| Flatten | — | $(\text{batch\_size}, \text{window\_size}/8 \times 112)$ |
| Dense | neurons: 64, activation: elu | $(\text{batch\_size}, 64)$ |
| Dense | neurons: 2, activation: linear | $(\text{batch\_size}, 2)$ |

Table 9: Architecture of the "functional body" neural network. This model has $1,157,394$ trainable parameters, amounting to $4.42\,\text{MB}$.

## B  Details on the SpatialFilterCNN

A spatial filter, in contrast to temporal filters, is a filter that acts across different electrodes, combining signals at a single time point. Fuhl et al. (2023) proposed a neural network architecture, where the first layer shall act as a (learnable) spatial filter, for the EEGEyeNet data. We refer to this model as *SpatialFilterCNN*.

The model consists of a spatial filtering layer, two residual blocks and two fully connected layers at the output. The detailed architecture of the SpatialFilterCNN model is shown in Figure 3. In case of the EEGEyeNet data, the input is a matrix of shape $500 \times 128$ (time points $\times$ channels). The spatial filtering layer is a one-dimensional convolutional layer with 16 filters and a kernel size of 1. It is followed by a batch normalization layer and a ReLU activation function. The output of this layer is then passed through two residual blocks. Each residual block consists of two one-dimensional convolutional layers, with either 32 filters for the first residual block or 64 filters for the second. The first convolutional layer in each residual

**Architecture #3: Minimally Functional**

| Layer | Parameters | Output Shape |
|---|---|---|
| STEM | — | $(\text{batch\_size}, \text{window\_size}/2, 64)$ |
| ResBlock | filters: 64 | $(\text{batch\_size}, \text{window\_size}/2, 64)$ |
| ResBlock | filters: 64 | $(\text{batch\_size}, \text{window\_size}/2, 64)$ |
| AvgPool | pool_size: 2, strides: 2 | $(\text{batch\_size}, \text{window\_size}/4, 64)$ |
| ResBlock | filters: 112 | $(\text{batch\_size}, \text{window\_size}/4, 112)$ |
| ResBlock | filters: 112 | $(\text{batch\_size}, \text{window\_size}/4, 112)$ |
| AvgPool | pool_size: 2, strides: 2 | $(\text{batch\_size}, \text{window\_size}/8, 112)$ |
| FuncDense | neurons: 512, activation: elu, n_functions: 12, basis_type: Legendre, pooling: True | $(\text{batch\_size}, 512)$ |
| Dense | neurons: 512, activation: elu | $(\text{batch\_size}, 512)$ |
| Dense | neurons: 2, activation: linear | $(\text{batch\_size}, 2)$ |

Table 10: Architecture of the "minimally functional" neural network. This model has $1,275,570$ trainable parameters, amounting to $4.87\,\text{MB}$.

block has a kernel size of 9, and uses padding to keep the first dimension of the output shape the same as the input shape. This is followed by a Batch Normalization layer and a ReLU activation function.

The second convolutional layer has a kernel size of 1 and is followed by another Batch Normalization layer. The output of the second convolutional layer is then added to the input of the residual block, which is passed through another one-dimensional convolutional layer in order to match the output shape of the second convolutional layer. The added output is then passed through a ReLU activation function and an average pooling layer with a pool size of 2 and a stride of 2. After the second residual block the output is flattened and passed through a fully connected layer with 256 neurons and a ReLU activation function. The output of this layer is then passed through another fully connected layer with 2 neurons, which is the output of the model and represents the prediction of the gaze position in pixels.

The hyperparameters of the SpatialFilterCNN are given in Table 11.

| Hyperparameter | Range / Setting |
|---|---|
| Window size | 128 to 1024 samples |
| Learning rate | $10^{-5}$ to $10^{-1}$ |
| Spatial filtering | Enabled or Disabled |
| Equally sized kernels | Enabled or Disabled |
| $N_S$ (number of spatial filters) | 4 to 64 |
| $N_1$ (filters in first residual block) | 8 to 128 |
| $N_2$ (filters in second residual block) | 16 to 256 |

Table 11: Hyperparameter ranges explored.

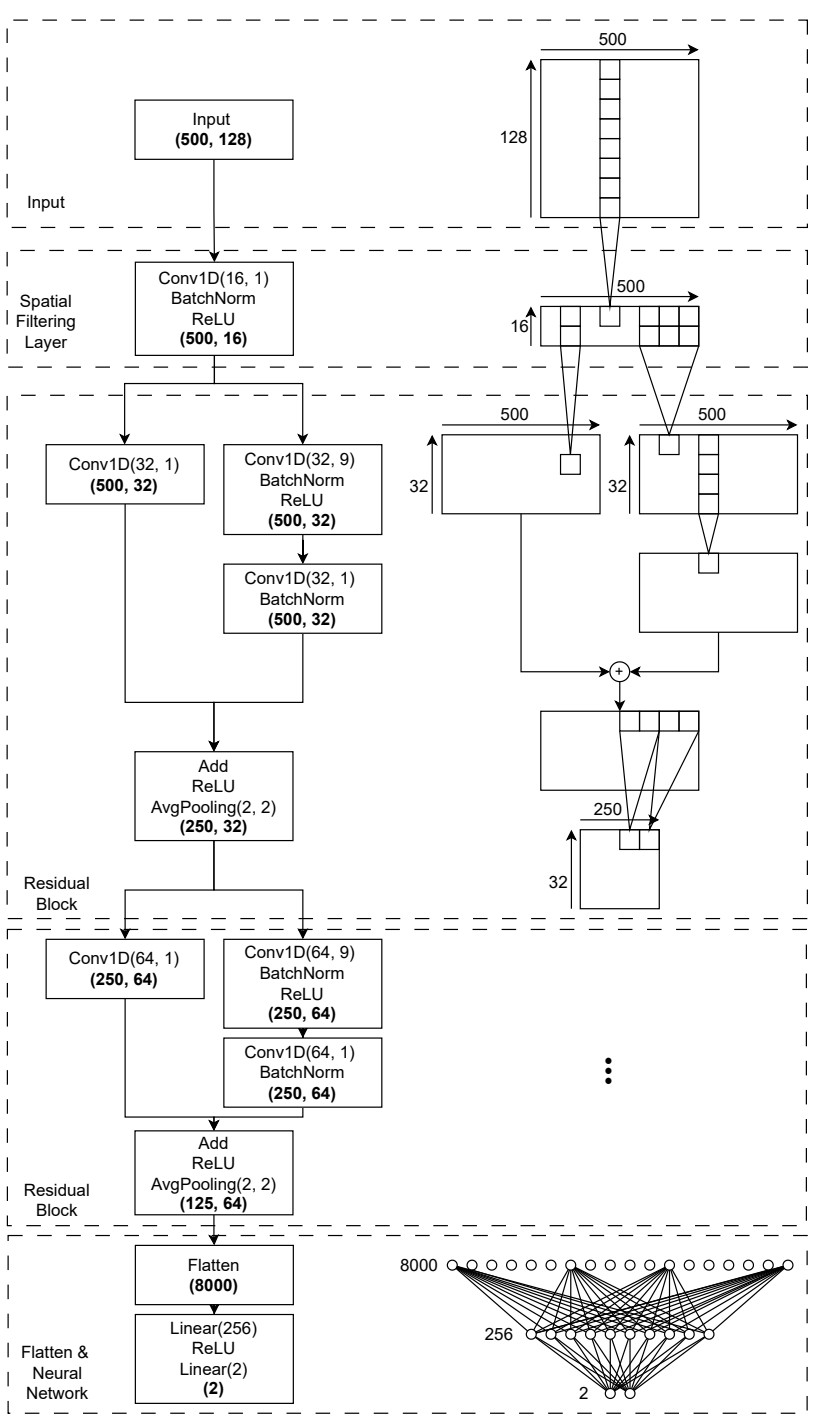

Figure 3: The architecture of the SpatialFilterCNN model, which on a high level consists of the input, a spatial filtering layer, two residual blocks and a fully connected neural network at the output. Conv1D($n$, $m$) denotes a one-dimensional convolutional layer with $n$ filters and a kernel size of $m$, AvgPooling($n$, $m$) average pooling with pool size $n$ and stride $m$ and Linear($n$) a layer of $n$ fully connected neurons.

