# OpenReview forum: "EEG-EyeTrack: A Benchmark for Time Series and Functional Data Analysis with Open Challenges and Baselines"
_TMLR — Rejected by TMLR_

### Review · Reviewer_YEYQ · 2025-04-28

**Summary Of Contributions:**

The authors curate a new EEG-based Eye-Tracking dataset and position it as a realistic test-bed for time-series/functional-data methods.

The evaluation protocol is well defined, across different challenges.

Several baselines like FNN are benchmarked on the new dataset and on EEGEyeNet.

**Audience:**

Yes

**Broader Impact Concerns:**

The dataset contains neuro-physiological signals that can reveal cognitive or health traits. The manuscript should detail anonymisation steps, data-sharing restrictions, and explicitly caution against potential misuse for covert attention tracking or surveillance. No such discussion is present.

**Claims And Evidence:**

Yes

**Requested Changes:**

The authors are suggested to tune FNN hyper-parameters.

Provide 5-run means with std and paired statistical tests.

Add at least one classical FDA model like FPCA + linear regression and a sequence model (e.g., LSTM, transformer) to anchor performance.

Include participant demographics, recording environment, and ethics/consent statement; clarify license

**Strengths And Weaknesses:**

Strength:

First consumer-grade EEG+gaze benchmark;

The paper is well organized;

Implements and documents strong CNN baseline; ablation between functional vs. conventional layers is insightful

Provides multiple metrics, qualitative plots, and ablation

Weakness:

Only ~12h data & 4 electrodes, which may limit representativeness

Ethical and demographic information is minimal; licensing of data/code not stated

Hyper-parameters tuned only for CNN, not FNNs

No other baselines like ridge regression, LSTM, transformer etc

No statistical significance tests in the reported results

---

> ### Author Response · Authors · 2025-06-20
>
> We thank the reviewer for their overall positive assessment of our work and for the helpful feedback. We hope to address all of their questions and concerns below.
>
> First, we want to clarify that we do not present a new dataset. As pointed out by another reviewer, our main contribution is **the benchmarking and task formalization based on the consumer-grade dataset**. The dataset itself was presented previously accompanied by a data descriptor by \[1\]. We added a comment to Section 3, referring to the original data descriptor, which contains additional information about the participants' demographics, ethics/consent statement etc. Additionally, we address potential misuse.
>
> In the following, we address the mentioned weaknesses (W) and requested changes (RC).
>
> W1: Only \~12h data & 4 electrodes, which may limit representativeness
>
> A1: While we agree that generally "more data is better", the given dataset is of a similar order as other datasets containing eye movement and EEG data. For example, the EEGEyeNet \[2\], contains approximately 48 hours from 356 participants, but only less than 8 hours of saccadic movement on a grid and approx. 1.5h of free eye movement, where the latter is specifically interesting from a functional data analysis (FDA) point of view. The "RaCCooNS" dataset includes recordings from 37 participants reading 200 sentences each \[3\]. While the total duration of the dataset is not provided, a total duration of 9 hours (~15 min per participant) is a reasonable guess. With approx. 12 hours of EEG recording from 113 participants, the considered dataset is of a similar order as the others. The reduced number of electrodes is due to the use of consumer-grade hardware, which we believe to be a strength of the selected dataset.
>
> W2: Ethical and demographic information is minimal; licensing of data/code not stated
>
> A2: As clarified above, we do not present the dataset itself, but the benchmarking and task formalization. We added a comment on ethical and demographic information, and refer to the original work for additional information. Also, we added that our own code is licensed under the MIT license.
>
> W3: Hyper-parameters tuned only for CNN, not FNNs
>
> A3: Thanks for pointing this out. The missing hyperparameter tuning is indeed a major shortcoming. We are currently working on the hyperparameter tuning for FNNs and will report results analogous to the CNNs in the final version of the manuscript. However, the training takes a considerable time, and we do not know yet, if we can present our new results before the discussion period ends.
>
> W4: No other baselines like ridge regression, LSTM, transformer etc
>
> A4: We agree that additional baseline models strengthen our benchmark experiments. Once we are done with hyperparameter tuning, we will additionally conduct experiments using the suggested baselines (FPCA + Linear Regression, LSTM).
>
> W5: No statistical significance tests in the reported results
>
> A5: Thanks for mentioning this point. Indeed additional error measures, such as the mean and standard deviation of multiple runs, allow conclusions about the generalization of models. Once we are done with hyperparameter tuning, we will train each considered model five times and report means and standard deviations for the dataset from \[1\], analogously to the results for the EEGEyeNet in Table 4. For statistical tests, though, we believe that the sample size $n=5$ is too small for conclusive non-parametric tests.
>
> RC6: The authors are suggested to tune FNN hyper-parameters.
>
> A6: We are currently working in hyperparameter tuning for FNNs (see A3)
>
> RC7: Provide 5-run means with std and paired statistical tests.
>
> A7: We are currently working on the experiments (see A5)
>
> RC8: Add at least one classical FDA model like FPCA + linear regression and a sequence model (e.g., LSTM, transformer) to anchor performance.
>
> A8: We are currently working on the baseline models FPCA + Linear Regression and LSTM (see A4). Results will be added to the final version of the manuscript.
>
>
>
> References
>
> \[1\] Afonso, T. V., & Heinrichs, F. (2025). Consumer-grade EEG-based eye tracking. _arXiv preprint arXiv:2503.14322_.
>
> \[2\] Kastrati, A., Płomecka, M. B., Pascual, D., Wolf, L., Gillioz, V., Wattenhofer, R., & Langer, N. (2021). EEGEyeNet: a simultaneous electroencephalography and eye-tracking dataset and benchmark for eye movement prediction. . In J. Vanschoren and S. Yeung (eds.), *Proceedings of the Neural Information Processing Systems Track on Datasets and Benchmarks*, volume 1. Curran, 2021.
>
> \[3\] Frank, S. L., & Aumeistere, A. (2024). An eye-tracking-with-EEG coregistration corpus of narrative sentences. _Language Resources and Evaluation_, _58_(2), 641-657.

---

### Review · Reviewer_1Xb4 · 2025-06-06

**Summary Of Contributions:**

The paper proposes new benchmark/challenges for functional data analysis derived from the publicly available "Consumer-Grade EEG and Eye-Tracking Dataset", it provides baseline comparisons using functional neural networks and conventional models for the task of eye movement reconstruction.

**Audience:**

Yes

**Broader Impact Concerns:**

The reviewer did not identify major broader impact concerns.

**Claims And Evidence:**

No

**Requested Changes:**

I will summarise my previous comment in actionable points in what follows:

I suggest clearly delineating that the dataset is reused from prior work and better motivating the specific novel contributions of this paper. This took some time for me to realize.

It would be beneficial to provide statistics on missing data and justify the use of SARIMA for imputation over other possible methods. Similarly also please explain the participant exclusion impact.

I have previously indicated my concerns on the exclusion of pooling and the reliance on the smoothness assumption. I advise analyzing how the proposed FNN handles local noise in the absence of pooling layers. Include or extend an ablation study to explore this trade-off.

The authors acknowledge that the results are mixed, however, more analysis or explanations are required to give value to the work.

Minor reformatting could improve reading, such as removing bullet lists in favour of tables or appendices.

**Strengths And Weaknesses:**

The proposed testbed for FFNs is interesting, and it is provided in the form of a challenges defined on a publicly available dataset, specifically tailored for FDA tasks. However, I have a concern that its contribution is limited as the dataset itself is not novel contribution. The emphasis on FDA and eye movement reconstruction is commendable, but the motivation feels overstated, given the narrow scope of the study.

On a technical note, regarding the dataset usage, the authors employ SARIMA for imputing missing data, and I am concerned that it may risk introducing artificial signal patterns. Moreover, the exclusion of several participants could bias results. It would be great if clarification is provided this.

The diversity of FDA challenges proposed by the authors is valuable and valid, and the experimental evaluations are thorough in the scope of the proposed challenges.

I have some concerns on the functional neural network design: I understand that it omits pooling to preserve smoothness, raising my concerns about its robustness to local noise / signal variations commonly present in EEG or eye-tracking data. The trade-offs between smoothness, generalization, and overfitting is unclear and might benefit from a more targeted ablation study.

I agree that results are extensive, especially on smooth movements, but overall conclusions are inconclusive, and the functional models do not consistently outperform traditional baselines across all settings. The mixed results may require additional investigation or at least a more in depth explanation.

On a very minor note, the presentation can be improved by condensing lengthy bullet lists of hyperparameters into tables or relegating them to an appendix.

---

> ### Author Response · Authors · 2025-06-20
>
> We thank the reviewer for their helpful feedback and hope to address all of their questions and concerns below. Based on the feedback, substantial parts of the manuscript have been modified and added. Changes to the manuscript are highlighted in red, so that they can be identified easily.
>
> The experiments suggested by Reviewer YEYQ are currently being prepared and conducted. Once available, the corresponding results will be presented in the final version of the manuscript. Since the training takes a substantial amount of time, we do not know yet, if we can present our new results before the discussion period ends.
>
> In the following, we answer to the requested changes (RC) that address the mentioned weaknesses and concerns.
>
> RC1: I suggest clearly delineating that the dataset is reused from prior work and better motivating the specific novel contributions of this paper. This took some time for me to realize.
>
> A1: We have adapted the abstract and introduction to emphasize that the dataset has already been published. Additionally, we added remarks to Section 4 (Data and Open Challenges, formerly Section 3).
>
> RC2: It would be beneficial to provide statistics on missing data and justify the use of SARIMA for imputation over other possible methods. Similarly also please explain the participant exclusion impact.
>
> A2: Thank you for this suggestion. We have added a new paragraph on missing data and SARIMA imputation, jointly with a summary table. More specifically, we state the missing-data rates per channel (between 0.0\,\% and 24.1\,\%, median 6.3\,\%) and the number of recordings exceeding 10\% of missing data (27 out of 407 recordings).
> We chose to retain the SARIMA model for missing value imputation proposed by \[1\]. Through this choice, we want to ensure comparability with other work based on the data. The SARIMA model was selected by \[1\] the author of the data descriptor to explicitly model the 50Hz background noise.
> Regarding participant exclusion, only recordings with technical errors, i.\,e., hardware failures, were dropped. This process is uncorrelated with any participant characteristics (which are anonymized), making systematic bias unlikely.
>
> RC3: I have previously indicated my concerns on the exclusion of pooling and the reliance on the smoothness assumption. I advise analyzing how the proposed FNN handles local noise in the absence of pooling layers. Include or extend an ablation study to explore this trade-off.
>
> A3: We added a new section on functional data analysis, including definitions of functional neurons and functional convolutional layers. In this section, we added a comment on smoothing, hence filtering out local noise. In particular, if the (functional) weight of a convolution is differentiable, and the input of the convolution is $k$-times differentiable, the output of the convolution will be $(k+1)$-times differentiable. If the input is only bounded, the output will be continuous; if the input is continuous, the output will be differentiable. Deep functional neural networks (containing multiple functional convolutional layer) smooth out local noise.
> As stated before, we are currently tuning hyperparameters of the functional neural networks. Once we have tuned those parameters, we will conduct an ablation study with different levels of local noise. The results will be added to the final version of the manuscript, but possibly only after the discussion period ends.
>
> RC4: The authors acknowledge that the results are mixed, however, more analysis or explanations are required to give value to the work.
>
> A4: Comparing FNNs with the conventional SpatialFilterCNN (with tuned hyperparameters) is currently ill-posed. Once the hyperparameters for FNNs are tuned and results are available, we will revise the comparison and the subsequent conclusions. Generally, though, FNNs seem favorable over the SpatialFilterCNN, as discussed in Section 7 (formerly 6) and in the conclusion (Section 8). Especially in the setting "level-2-smooth", which is the most interesting from a FDA point-of-view, the functional neural networks are clearly favorable over their conventional counterparts and yield results similar to the tuned SpatialFilterCNN, but with substantially lower (= better) precisions.
> We will update the discussion and conclusions once the new results are available.
>
> RC5: Minor reformatting could improve reading, such as removing bullet lists in favour of tables or appendices.
>
> A5: As suggested, we have replaced the list with a table and moved it to the appendix.
>
>
> References
>
> \[1\] Afonso, T. V., & Heinrichs, F. (2025). Consumer-grade EEG-based eye tracking. _arXiv preprint arXiv:2503.14322_.

---

### Review · Reviewer_K9zB · 2025-06-11

**Summary Of Contributions:**

This study repurposes the previously released Consumer-Grade EEG-based Eye Tracking dataset as a benchmark for functional data analysis (FDA) and time series modeling, particularly for predicting eye movements from EEG signals. The authors define a series of open challenges—including gaze prediction, classification, clustering, and change point detection—and propose standardized evaluation metrics, with a focus on Mean Euclidean Distance (MED).

To establish performance baselines, the paper introduces and evaluates several Functional Neural Network (FNN) architectures, varying in the degree of functional layer integration (fully functional, functional body, minimally functional). These are benchmarked against conventional CNNs and the SpatialFilterCNN model, using both the consumer-grade dataset and the research-grade EEGEyeNet dataset.

The study finds that FNNs offer smoother and often more consistent predictions, particularly for smooth eye movements, though their performance is sensitive to the choice of architecture and dataset. The work positions itself as a step toward FDA-oriented benchmarks for EEG-based modeling, while also contributing initial architectural insights into functional neural networks for brain-signal-to-behavior prediction.

**Audience:**

Yes

**Broader Impact Concerns:**

No ethical concerns are identified. A Broader Impact Statement does not appear to be necessary for this work.

**Claims And Evidence:**

No

**Requested Changes:**

- Please provide appropriate citations when introducing key concepts in the introduction, such as *functional data analysis (FDA)* and *functional neural networks (FNNs)*, to better ground them in the introduction.

- Consider elaborating on important terms such as *function-on-function* and *scalar-on-function regression*. The current descriptions are somewhat brief and may be confusing to readers unfamiliar with these FDA-specific formulations.

- I recommend including a diagram of the study framework or model architecture in the main text, rather than placing it in the appendix. A visual overview would significantly enhance clarity and accessibility for the reader.

- Clarify the overall scope and positioning of the paper. It remains unclear whether the main contribution lies in the methodological development of FNNs for EEG-to-gaze modeling, or in the benchmarking and task formalization based on the consumer-grade dataset. If the paper aims to do both, a clearer structure and separation of these contributions is needed to avoid ambiguity.

**Strengths And Weaknesses:**

Strengths:
- EEG-based eye tracking is an emerging research direction, especially in low-cost and mobile brain-computer interfaces (BCIs). The paper addresses this timely need by evaluating models on consumer-grade EEG data.
- While the dataset is not new, the paper adds value by re-framing it as a benchmark for functional data analysis (FDA) and time series modeling, complete with task definitions and evaluation metrics.
- The authors conduct detailed experiments comparing FNNs and SpatialFilterCNNs across both consumer-grade and research-grade datasets, including ablation studies with functional vs. standard layers.

Weaknesses:
- When introducing the foundational concepts underlying the methods and models used in this paper, there is a significant lack of key references to relevant prior work.

- The major concern lies in the paper’s overly broad and somewhat unfocused scope. While it offers interesting insights into EEG-based gaze prediction using functional neural networks, it simultaneously attempts to define a benchmark, explore multiple open challenges, and propose architectural comparisons. However, none of these components are explored in sufficient depth to fully realize their potential, which may cause confusion and disrupt the reader’s understanding. A clearer separation between the benchmark framing and the methodological contributions would significantly enhance the clarity and impact of the work.

- Despite the framing, the dataset used in this study is not newly introduced. It was previously published (Afonso & Heinrichs, 2025), which limits the novelty of the benchmark contribution.

- Several benchmark tasks (e.g., participant classification, change point detection) are mentioned as potential research challenges but are not empirically validated in this work. As a result, the benchmark remains largely conceptual rather than being supported by practical and systematic evaluations.

---

> ### Author Response · Authors · 2025-06-20
>
> We thank the reviewer for their helpful feedback and hope to address all of their questions and concerns below. Substantial parts of the manuscript have been modified and added. Changes to the manuscript are marked in red.
>
> First, note that the experiments suggested by Reviewer YEYQ are currently being prepared and conducted. The results will be presented in the final version of the manuscript. However, since the training requires a considerable amount of time, it remains unclear whether we will be able to include the results before the discussion period ends.
>
> In the following, we answer to the requested changes (RC) that address the mentioned weaknesses.
>
> RC1: Please provide appropriate citations when introducing key concepts in the introduction, such as _functional data analysis (FDA)_ and _functional neural networks (FNNs)_, to better ground them in the introduction.
>
> A1: We have added a brief introduction to functional data analysis, along with relevant references, to the introduction.
>
> RC2: Consider elaborating on important terms such as _function-on-function_ and _scalar-on-function regression_. The current descriptions are somewhat brief and may be confusing to readers unfamiliar with these FDA-specific formulations.
>
> A2: An entire section devoted to the main concepts of FDA has been added. This section provides details on scalar- and function-on-function regression and foundations of functional neural networks.
>
> RC3: I recommend including a diagram of the study framework or model architecture in the main text, rather than placing it in the appendix. A visual overview would significantly enhance clarity and accessibility for the reader.
>
> A3: A figure containing the "benchmark pipeline" and the general architecture of the FNNs has been added.
>
> RC4: Clarify the overall scope and positioning of the paper. It remains unclear whether the main contribution lies in the methodological development of FNNs for EEG-to-gaze modeling, or in the benchmarking and task formalization based on the consumer-grade dataset. If the paper aims to do both, a clearer structure and separation of these contributions is needed to avoid ambiguity.
>
> A4: The main contribution of the manuscript is the benchmarking and task formalization, establishing the dataset as a new benchmark in FDA. The methodological development of FNNs for EEG-to-gaze reconstruction serves as a secondary outcome, providing initial baseline results. The bullet points at the end of the introduction have been modified to clarify the main contribution. Additionally, Section 4.1 (formerly 3.1) has been restructured.

---

### Review · Reviewer_cTqG · 2025-06-17

**Summary Of Contributions:**

The paper proposes a more challenging EEG eye-tracking benchmark dataset for functional neural networks. A series of experiments are conducted to evaluate the effectiveness of FNNs on this dataset and compared with the baselines.

**Audience:**

Yes

**Claims And Evidence:**

No

**Requested Changes:**

- I would suggest the authors to redefine the premise of the paper and make the contributions clear.

- Add more experiments to highlight the advantages and limitations of FNNs compared to the baselines and make conclusive recommendations based on those results

**Strengths And Weaknesses:**

EEG-based eye tracking is becoming an important problem since it is the most promising in terms of realizing an actual BCI. Hence, large-scale datasets that can serve as benchmark for different EEG decoding methods can be very useful. However, the paper falls short of realizing that promise due to the following issues:

- My main concern is the confusing premise of the paper. On one hand, it claims to propose a new dataset and metrics for EEG based eye tracking (with the motivation of advancing EEG based BCIs). One the other hand, it claims to provide a challenging benchmark for FNNs which so far have been evaluated on trivial datasets. The issue for both these claims is that the dataset is not new and hence cannot be claimed as a contribution of the paper. Infact, it is not even clear from the writeup that the dataset is not new. To a naive reader, this could easily come across as a dataset that is proposed by the paper.

- There are not enough experiements and discussion on the EEG aspect to make this an EEG-centric paper. This would entail discovering and discussing what are the EEG features for gaze tracking. This implies that the crux of this paper is to evaluate the FNNs against standard methods on a challenging method (which is a fair contribution but the paper needs to be written that way and discussion should reflect it).

- Even taking the point that the paper wants to analyze FNNs, I am not sure if the conclusions are clear. From the results, FNNs do not seem to have a clear advantage over the baselines. The baselines themselves are not representative of the literature. I am not sure what is the key takeaway message.

---

> ### Author Response · Authors · 2025-06-20
>
> We thank the reviewer for their helpful feedback and hope to address all of their concerns below. The manuscript has been modified substantially, with changes marked in red.
>
> In the following, we answer to the requested changes (RC) that address the mentioned weaknesses.
>
> RC1: I would suggest the authors to redefine the premise of the paper and make the contributions clear.
>
> A1: We thank the reviewer for pointing out this issue. In fact, the confusing scope has been mentioned by other reviewers as well. The main contribution of the manuscript is the **benchmarking and task formalization**, establishing the dataset as a new benchmark in FDA. While the dataset itself has been presented with an accompanying data descriptor (see \[1\]), no aim for its analysis has been specified, nor any quantitative results have been given. We formalize the task of gaze reconstruction from EEG data and specify evaluation metrics, establishing the dataset as a benchmark in FDA. Additionally, we provide baseline results, using different models, for the task of scalar-on-function regression. The manuscript is intended to be a starting point for the evaluation of future developments in FDA, providing solid baseline results.
>
> We have adapted the manuscript to clearly state our contribution. In particular, we clarified that the dataset has already been published by \[1\].
>
> RC2: Add more experiments to highlight the advantages and limitations of FNNs compared to the baselines and make conclusive recommendations based on those results
>
> A2: Thanks for the suggestion, we are currently working on multiple experiments, as proposed by other reviewers. More specifically, we are currently:
> - Tuning hyperparameters of FNNs (for better comparability with the conventional SpatialFilterCNN).
> - Training the final (tuned) FNNs, the previously used baseline models (control models of FNNs, SpatialFilterCNN) and new baseline models (FPCA + Linear Regression, LSTM) multiple times. Based on these multiple runs, we will provide means and standard deviations of the results.
> - Preparing an ablation study to investigate how functional and non-functional models handle local noise.
> These experiments are currently running/being prepared and we do not know yet, if we can include them into the manuscript before the discussion period ends. However, the additional results, and conclusions and suggestions based on these results will be added to the final version of the manuscript.
>
>
>
> References
>
> \[1\] Afonso, T. V., & Heinrichs, F. (2025). Consumer-grade EEG-based eye tracking. _arXiv preprint arXiv:2503.14322_.

---

### Decision · Action_Editor_vXnm · 2025-08-04

**Recommendation:** Reject

**Additional Comments:**

The authors need to address the technical points raised above and clarify the core contribution of this work.

**Audience:**

Yes

**Audience Explanation:**

There is a clear consensus among reviewers that benchmarks are needed, and that there is value in the proposed reconstruction of eye movements from EEG data. The use of existing data makes it less clear what the real contribution is (see the comment above).

**Claims And Evidence:**

No

**Claims Explanation:**

The paper has not yet convinced the reviewers for the following technical reasons:
* Hyper-parameter tuning: The FNN experiments lack hyper-parameter tuning, making the work appear unfinished and defeating the purpose.
* Additional baselines are needed for this kind of work.
* Error bars and statistical conclusions are still missing to strengthen the paper's main messages.

Overall, the objective should be clarified, as the paper does not seem to do a good job of outlining the strengths of FDA techniques. Is the aim to establish a new benchmark or demonstrate the effectiveness of FDA methods?
Since the data are not new, it would be beneficial to focus on the advantages of FDA techniques, but this would require additional work.

**Resubmission Of Major Revision:**

The authors may consider submitting a major revision at a later time.